

# Comparison of the LEO and CPMA-SP2 techniques for black-carbon mixing-state measurements

Arash Naseri ©[1], Joel .C Corbin ©[2], and Jason .S Olfert ©[1]

[1]Department of Mechanical Engineering, University of Alberta, Edmonton, Alberta, Canada
[2]Metrology Research Centre, National Research Council Canada, Ottawa, Ontario, Canada

**Correspondence:** Jason Olfert (jolfert@ualberta.ca, Office Tel: 780-492-2341)

**Abstract.** It is necessary to measure the mixing states of light-absorbing carbon (LAC) particles to reduce uncertainties in climate forcing due to particulate from wildfires and biomass combustion. For refractory LAC (normally called refractory black carbon; rBC), such measurements can be made using the single particle soot photometer (SP2). The SP2 measures the incandescent mass of individual particles due to heating by a 1064 nm laser. The SP2 also monitors single-particle light scattering

from rBC plus internally mixed material (*e.g., coatings* of volatile particulate matter). rBC mixing states can be estimated from SP2 measurements by combining the scattering and incandescence signals. This is the basis of the published methods known as the (*i*) scattering–incandescence lag-time, (*ii*) leading-edge only (LEO), and (*iii*) normalized derivative methods. More recently, the tandem centrifugal particle mass analyzer (CPMA)–SP2 method has been developed. The CPMA–SP2 method does not rely on the SP2 scattering signals and, therefore truly measures the rBC mass fraction, with no assumptions regarding particle

composition or morphology. In this study, we provide the first quantitative comparison of the light-scattering and CPMA–SP2 methods for measuring mixing state. We discuss the upper and lower limits of detection (in terms of both rBC and coatings), temporal resolution, role of counting statistics, and errors associated with the measurements. We use a data set of atmospheric particles sampled at a regional background site (Kamloops, Canada; about 350 km northeast of Vancouver, British Columbia, Canada), where the majority of rBC was emitted by seasonal wildfires. In the overall comparison among measurement meth-

ods, the CPMA–SP2 method is found to have significantly better systematic uncertainties than the light-scattering methods for wildfire smoke. For example, the light-scattering methods could not quantify coatings on half of the rBC particles, because their light-scattering signals were below the SP2 detection limit. Consequently, the bias in SP2-only estimates of rBC mixing states depends on the size distribution of the rBC particles. Although more accurate, CPMA–SP2 measurements require significantly more time to acquire, whereas SP2-only light-scattering analyses (both LEO and lag-time) can provide near real-time

qualitative information representing large rBC particles.

## 1 Introduction

Atmospheric light-absorbing carbon (LAC) plays an important role in the Earth's radiative balance, affecting the amount of terrestrial and solar radiation absorbed by the atmosphere. This affects the Earth's surface temperatures as well as precipitation patterns (Samset, 2022). The short lifetime of LAC, as well as its toxicity in the human lung, makes it an attractive target for



short-term climate mitigation (Grieshop et al., 2009; Shindell et al., 2012). A large fraction of atmospheric LAC is emitted by wildfires and biomass burning (Bond et al., 2013). These sources release LAC in the form of black carbon (BC, or soot; insoluble aggregates of partially graphitized carbon), brown carbon (BrC; soluble light-absorbing organic molecules including humic-like substances and polycyclic aromatic hydrocarbons), and tarballs (TBs; insoluble, amorphous-carbon spheres) (see Corbin et al., 2019b; Michelsen et al., 2020, for a detailed discussion of these categories). While BC is the best known and often

the most abundant of these LAC types, BrC has received increasing attention (Laskin et al., 2015) and recent work suggests that TBs may be the dominant LAC species in some wildfire (Adachi et al., 2019; Chakrabarty et al., 2023) and marine-engine smoke (Corbin et al., 2018).

Major gaps remain in our understanding and prediction of atmospheric light by LAC. The relative abundance of BC, BrC, and TBs is one major question, which must be addressed using measurement techniques capable of quantifying these species

separately (e.g. Laskin et al., 2015; Adler et al., 2019; Corbin and Gysel-Beer, 2019). Another major question is the degree to which light absorption by these species is enhanced by their internal mixing with non-absorbing species such as organic and inorganic matter or water. In some cases, such internal mixing can double the resulting light absorption of the mixed LAC particle (Cappa et al., 2019). Detailed modelling has showed that a size-resolved understanding of internal mixing is essential to explain the existing variety of observations (Fierce et al., 2017, 2020). A quantitative answer to the mixing-state question,

therefore, requires quantitative measurements of the size-resolved mixing state of atmospheric BC.

One technique capable of providing such size-resolved mixing state measurements is the single-particle soot photometer (SP2, Schwarz et al., 2006; Stephens et al., 2003). The SP2 measures the incandescence and scattering of individual particles during exposure to a high-intensity, continuous-wave, 1064 nm laser. The vast majority of incandescence in atmospheric particles is due to BC (Schwarz et al., 2006), although it should be noted that both anthropogenic iron particles (Moteki et al., 2017)

and marine-engine TBs (Corbin and Gysel-Beer, 2019), as well as laboratory TB surrogates (Sedlacek III et al., 2018) have been shown to generate detectable rBC signals in the SP2. On the other hand, scattering signals are produced by all materials present in sufficient quantity within a particle. Therefore, a comparison of the SP2 scattering and incandescence signals allows for an estimate of mixing states. This has been done via the complex *leading-edge only* (LEO) technique (Gao et al., 2007) (which extrapolates from the initial light-scattering signals of a particle, to avoid issues of coating evaporation) as well as the

conceptually similar normalized derivative method (Moteki and Kondo, 2008). Below, we refer only to the SP2–LEO method, although similar results are expected from the normalized derivative method. A distinct method of SP2 light-scattering analysis, the simplistic lag-time analysis (which simply compares the time at which scattering and incandescence peak) has often been used. Lag-time analysis is much simpler, categorizing particles as *thickly-coated* if their scattering signal peaks before the incandescence signal, and *moderately-to-thinly coated* otherwise. These methods are discussed in more detail below.

The SP2–LEO estimation of coating thickness is limited for two reasons. First, the accuracy of the implicit physical model is significantly limited (Liu et al., 2017): coated rBC particles are not core-shell spheres. Second, the range of response of the SP2 detectors does not span all relevant scenarios. For example, very small (but significant in mass concentration) rBC particles do not scatter enough light to be detected, so the otherwise broad range of response of the SP2 incandescence detector cannot be fully exploited.



A significant improvement to the accuracy of SP2-based mixing state measurements is achieved by combining the SP2 in tandem with the centrifugal particle mass analyzer (CPMA, Olfert and Collings, 2005). CPMA–SP2 data provide quantitative, two-dimensional distributions of the mass fraction of rBC once multiple charging effects are removed (Section 3.2). The CPMA–SP2 technique avoids the two SP2–LEO limitations above, since it utilizes only the incandescence channel of the SP2 and requires no physical model for interpretation. These advantages were acknowledged in prior CPMA–SP2 work (Liu et al.,
2017; Broda et al., 2018), but prior work has not performed a quantitative comparison of SP2–LEO and CPMA–SP2 results. This gap exists in part because the appropriate inversion algorithms for CPMA–SP2 analysis have only recently been developed (Naseri et al., 2021a, b). Furthermore, Naseri and Olfert (2023) conducted a recent study that emphasizes the considerable significance of experimental characterization of the CPMA transfer function, which can significantly enhance the inversion of CPMA-SP2 data.

In this work, we discuss and quantify differences in the SP2–LEO and CPMA–SP2 methods in detail. Specifically, we compare the LEO method, incandescence lag-time, and CPMA–SP2 methods in terms of detection size range, temporal resolution, counting statistics, and associated uncertainties. We evaluate these methods using atmospheric measurements taken in Kamloops, British Columbia, Canada, during episodes dominated by predominately urban and highway soot particles (low coating content) or moderate to heavy wildfire smoke particulate (moderate to high coating content). Our results provide clear insights
into the limitations of the SP2–LEO method. Our data also suggest that not only coated BC, but also TBs are detectable by CPMA–SP2, a result that we discuss briefly here and will return to in future work.

## 2  Experimental methods

### 2.1  Measurement location

The measurements were carried out in an urban setting in the city of Kamloops, British Columbia, Canada ($50°39'58.4''$
N, $120°21'45.5''$ W). The site was located 2.2 km from an air quality station operated by the British Columbia Ministry of Environment and Climate Change and $\sim 0.7$ km from the Trans-Canada highway, which is a major corridor for heavy-duty diesel vehicles carrying freight. Experiments took place on July 21 and July 22, 2021; however, three periods of time are used in this work as examples to compare the measurement methods. The examples include periods where ambient particulate levels were at: *i*) Case I; low concentrations and mostly thinly coated rBC particles, presumably composed of mostly urban
and highway emissions and may also contain some wildfire smokes (July 21, 11:54 am to 1:44; PM2.5 concentrations of 1.7 to 4 $\mu$g/m$^3$ as measured at the nearby air quality station), *ii*) Case II; moderate concentrations and mixture of thinly and thickly coated rBC particles due to urban and highway emissions and smoke from nearby forest fires having bimodal particle size/mass distribution (July 22, 11:31 am to 12:41 pm; PM2.5 concentrations of $\sim 104$ to 81 $\mu$g/m$^3$), and iii) Case III; high concentrations and mostly thickly coated rBC due to wildfire smoke (July 22, 10:23 to 11:29 am; PM2.5 concentrations of 122
to 104 $\mu$g/m$^3$).



## 2.2 Measurement system

Figure 1 shows the method used to measure ambient particles. Ambient particles are drawn through $\sim 2.5$ m of silicone conductive tubing and an X-ray bipolar aerosol charger (Model 3088, TSI Inc). The particles were then directly sampled by the SP2 or classified by mass-to-charge ratio by the CPMA (Cambustion Ltd.) before being measured by SP2. The SP2 flow rate was maintained at 0.12 L/min during the experiments. However, to lessen particle diffusional losses (in the sampling line and the CPMA), a additional pump and critical orifice was used to maintain a flow of 1.5 L/min from the ambient inlet to the SP2 inlet.

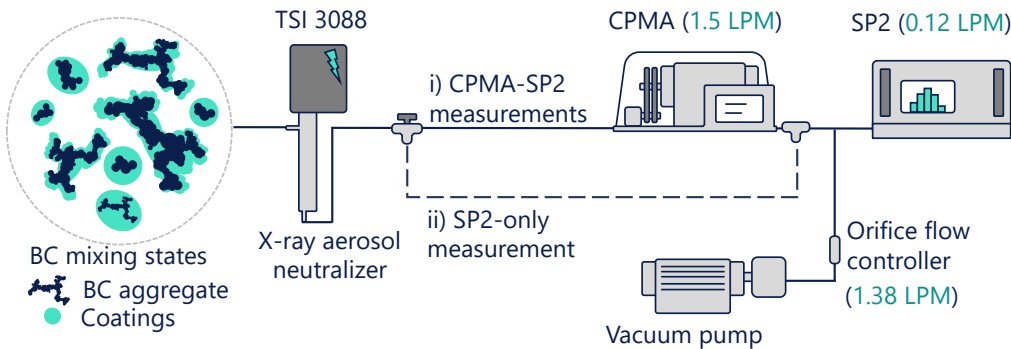

**Figure 1.** A schematic of the measurement system.

For CPMA–SP2 measurements, the CPMA was stepped through mass set points over the range of $m^* =$ 0.2–100 fg, with 5 CPMA set points per decade, and a CPMA resolution of 9. In general, 3 to 5 minutes were spent at each CPMA set point so that the total measurement time was limited to about an hour. This procedure led to about 1000 or more BC-containing particles being counted by the SP2 at each CPMA set point (the exception being CPMA set points near the limits of the distribution where particle counts were very low). These experimental settings (resolution, number of CPMA set points, SP2 counts) were chosen based on the optimized settings suggested by Naseri et al. (2021b) for the particle concentrations and measurement time frames in this work. The SP2-only measurements were taken at the beginning, end, and occasionally between CPMA–SP2 scans.

## 2.3 SP2 calibration

The SP2 incandescence signal was calibrated using soot from a miniature inverted soot generator (Argonaut Scientific Corp.; Kazemimanesh et al. (2019); Moallemi et al. (2019)) denuded with a catalytic stripper (Model CS08, Catalytic Instruments; Jacob and David (2010)). The denuded soot was classified with a CPMA over a wide range of set points and measured by the SP2, similar to the procedure used by Irwin et al. (2013). Since volatile material is removed from the rBC particles, the calibration particles have no coating by definition and the CPMA mass is equal to the rBC mass ($m_{\mathrm{p}} = m_{\mathrm{rBC}}$).[1] The scattering

---

[1] In this context, the masses measured for incandescing particles using this calibration data are to be interpreted as rBC equivalent mass.



signal was calibrated using polystyrene latex (PSL) sphere standards (Thermo Scientific 3000 Series) with a diameter of 300 nm. The PSI SP2 toolkit, version 4.114, was used to obtain the calibration curves for each signal.

## 3 Data analysis methods

### 3.1 SP2-only measurements

#### 3.1.1 Leading-edge only

In the SP2 literature, the amount of coating material in the rBC-containing particle is typically represented as a coating thickness ($t_{\text{coating}}$), which is calculated as

$$t_{\text{coating}} = \frac{d_{\text{opt}}(\text{RI}) - d_{\text{rBC}}}{2},$$
(1)

where, $d_{\text{opt}}(\text{RI})$ is the optical diameter of the particle (a function of particle volume, refractive index, and morphology), and $d_{\text{rBC}}$ is the mass-equivalent diameter of the rBC portion of the particle. The mass-equivalent diameter $d_{\text{rBC}}$ is simply the volume-equivalent diameter calculated from the SP2-measured rBC mass and a literature value of rBC material density (*i.e.,* $\rho_{\text{rBC}} = 1800 \text{ kg/m}^3$, Ouf et al. 2019):

$$d_{\text{rBC}} = \left(\frac{6m_{\text{rBC}}}{\pi \rho_{\text{rBC}}}\right)^{1/3}.$$
(2)

To estimate $d_{\text{opt}}$, the scattering cross-section is needed to be determined as an input to the Mie theory. The scattering cross-section of the particles, can be estimated from the peak intensity of the Gaussian scattering signal, which was calibrated with measurements of PSL of known size. There is, however, a confounding effect for scattering size measurements of rBC-containing particles, such that as coated rBC particles are heated by the laser beam, the radiatively heated rBC heats the coating by conduction, causing evaporation and resulting in a lower particle volume. A lower volume means a lower peak intensity of the scattering signal. Accordingly, the peak intensity of the scattering signal cannot be directly derived from the originally detected signal and requires the reconstruction of the undistorted scattering signal. Developed by Gao et al. (2007), the leading-edge-only (LEO) approach reconstructs the expected scattering signals of rBC-containing particles by fitting a Gaussian function to the part of the scattering signal that precedes the evaporation of the coating material, *i.e.,* the leading edge of the scattering signal. We defined the leading edge of the scattering signal as 3% of the maximum laser intensity based on scatter plots of LEO and standard analysis for non-absorbing particles, similar to previous studies (Gao et al., 2007; Taylor et al., 2015).

The LEO approach requires precise knowledge of the particle position within the SP2 laser beam. This knowledge is obtained using the so-called *split detector*. The split detector is a two- or four-part (depending on the SP2 model) avalanche photodiode, where the polarity on one half of the detector is inverted relative to the other half. The split detector is then oriented such that



when rBC-containing particles cross through the laser beam, the light scattered by them shifts from one element of the detector to the other. As particles move through the center of the detector, the scattering signal is split evenly, so that the measured scattering signal is zero. This process creates a clear notch that is used to infer the position of rBC-containing particles inside the laser beam, as the distance between the notch and the peak intensity of the laser changes only when the instrument optics are realigned during servicing (*i.e.,* it remains constant during the measurements).

Apart from the reconstructed scattering signal, several assumptions, like the refractive indices of the coating and rBC core, are required as inputs to Mie theory to obtain the optical diameter (scattering-equivalent size) of rBC-containing particles (Taylor et al., 2015). In reality, the exact value of rBC's refractive index is unknown and may vary from one rBC material to another; nonetheless, it has been empirically shown that $\kappa \approx n-1$, (Bond and Bergstrom, 2006; Moteki et al., 2010). Taking this constraint into account, the SP2 Toolkit uses a range of lookup tables containing scattering cross-sections at the corresponding

SP2 laser wavelength ($\lambda = 1064$ nm) for diverse core and coating refractive indices. The user then selects the most appropriate rBC refractive index by ensuring that the rBC volume predicted from scattering by uncoated rBC particles is similar to the rBC volume predicted from the incandescence signal. Uncoated rBC particles are defined from the light-scattering signal just prior to incandescence onset, when coatings are assumed to have evaporated. In our study, this resulted in $n_{\mathrm{rBC}} = 2.26 + i\,1.26$. It should be noted that this approach must not be mistakenly interpreted as quantifying the rBC refractive index, which requires

more supplementary measurements, but as a mechanism to ensure internal consistency in the LEO results.

The LEO approach is commonly employed to derive the size or mixing state of rBC-containing aerosols (Raatikainen et al., 2017; Sedlacek III et al., 2012; Zhang et al., 2018c, a). However, the scattering detection range for pure scattering particles (with no rBC portion) is narrow (*i.e.,* $\sim 200$–500 nm). The size range for LEO measurements of rBC-containing aerosols depends not only on the size of the particle, but also on the amount of rBC in the particle which affects the refractive index and

the evaporation rate of the coating portion.

To measure coating thickness/mass of rBC-containing particles using the LEO method, a spherical core-shell morphology with several core/shell property assumptions are required, namely, (*i*) a spherical void-free rBC core and values for: (*ii*) rBC material density ($\rho_{\mathrm{rBC}}$), (*iii*) coating material density ($\rho_{\mathrm{coating}}$), (*iv*) core refractive index ($n_{\mathrm{rBC}}$), and (*v*) coating refractive index ($n_{\mathrm{coating}}$). The following values are often used in the literature and are used in this work unless otherwise stated: $\rho_{\mathrm{rBC}}=$

1800 $\mathrm{kg/m^3}$ (Park et al., 2004; Bond and Bergstrom, 2006; Moteki et al., 2010; Corbin et al., 2018; Liu et al., 2020; Ouf et al., 2019), $\rho_{\mathrm{coating}} = 1000$ $\mathrm{kg/m^3}$ (Ditas et al., 2018; Liu et al., 2019), $n_{\mathrm{rBC}} = 2.26 + i\,1.26$ (Taylor et al., 2015; Moteki et al., 2010; Laborde et al., 2013; Zanatta et al., 2018; Dahlkötter et al., 2014), and $n_{\mathrm{coating}} = 1.5 + i\,0$ (Laborde et al., 2013; Nakayama et al., 2010; Liu et al., 2015; Yuan et al., 2021). It is well established that the core-shell assumption does not adequately capture the characteristics of all rBC-containing particles (Liu et al., 2017; Cappa et al., 2019), and the accuracy of

LEO in determining coating thickness relies on the four property values. The core/shell parameters utilized in LEO calculations are notably influenced by density and refractive index (Taylor et al., 2015), which play a crucial role in the accuracy of the results.



### 3.1.2 Normalized Derivative

The normalized derivative approach (Moteki and Kondo, 2008) in evaluating rBC mixing states is similar to the LEO approach
except a different methodology is used to estimate the undisturbed particle diameter. While a split-detector signal is used in
the LEO to derive the particle positional information, which is essential to recovering the undisturbed scattering signals, the
*normalized derivative* of the scattering signal is used in the ND approach to obtain analogous information. Thus, the ND
approach is similar to the LEO approach except that the detection limits and noise in the optical sizing of the coated particle
may be slightly different. Due to the high degree of similarity between ND and LEO models, the ND approach is not considered
in the present study.

### 3.1.3 Lag-time analysis

Lag-time analysis is a categorical approach for determining a general picture of the mixing state of BC. SP2 scattering signals
can have two peaks, with the second one occurring almost at the peak of the incandescence. While the second peak of the
scattering signal is typically higher than the first peak in thinly to moderately coated rBC particles, the opposite is true for
thickly coated rBC particles. For the latter, the global maximum of scattering signals typically occurs well before the peak
intensity of the incandescence signals, because of the much lower vapourization temperatures of typical coating material.
Thus, the time difference between the peak intensity of the time-resolved SP2 scattering and incandescence signals ($\Delta\tau = t_{\mathrm{max,scat}} - t_{\mathrm{max,incand}}$) is used to categorize rBC-containing particles into two groups: (*i*) rBC particles with *thick* coatings,
or (*ii*) *thin-to-moderate* coating (Schwarz et al., 2006; Moteki and Kondo, 2007; Subramanian et al., 2010; Corbin et al.,
2018). Generally, particles are categorized as thickly coated rBC if the time lag between the peak intensities of the scattering
and incandescence signals is more than $\sim 2~\mu$s, indicating considerable loss of coating material due to the heat absorbed
by the particles as they transverse the laser beam. For rBC particles with thin-to-moderate coating, the peak scattering and
incandescence signals occur nearly coincidentally (*e.g.,* time lag $< 2~\mu$s). The lag-time method cannot distinguish between
uncoated and thinly-to-moderately coated rBC particles.

One limitation of this method is that it cannot categorize rBC-containing particles whose rBC core and coating material are
fragmented by laser light (Moteki and Kondo, 2007; Sedlacek III et al., 2012; Moteki et al., 2014; Dahlkötter et al., 2014;
Sedlacek III et al., 2015). In such a case, the rBC portion reaches its boiling point and evaporates, while the unevaporated
coating portion is fragmented into smaller particles. A key to recognizing such a phenomenon is that the scattering signal
does not entirely vanish as the incandescence signal goes to zero. Secondly, the lag-time method may be confounded by
scenarios in which rBC forms within the SP2 laser (Sedlacek III et al., 2018; Corbin and Gysel-Beer, 2019). For more complex
categorizations, advanced methods should be implemented, *e.g.,* supervised machine learning method (Lamb, 2019) or more
complex analysis methods (Corbin and Gysel-Beer, 2019).

Depending on the rBC-containing particles measured and the SP2 used, the time lag threshold varies slightly. For instance,
Liu et al. (2022) and Zhang et al. (2018b) used the threshold of $\Delta\tau = 1.8~\mu$s and $1.6~\mu$s, respectively, while $\Delta\tau = 2.0~\mu$s
was used by Wang et al. (2014) and Wu et al. (2016). This variability may be caused by differences in coating composition,





SP2 flow rates, or laser powers. Nevertheless, this variation is almost negligible, *i.e.,* less than $\sim 0.4\,\mu$s (Moteki and Kondo, 2007), regardless of the atmospheric conditions and pollution level (Wu et al., 2016, 2017; Laborde et al., 2012). In the present study, we found that the low-gain channel yielded fewer invalid lagtime measurements, and therefore used this channel for our lag-time analysis. We used a time-lag threshold of $\Delta\tau = 2$ $\mu$s based on the observed bimodal distribution of lag times in our 210    data.

### 3.2    CPMA–SP2

The CPMA–SP2 system is used to measure the two-variable distribution of total particle mass $m_{\mathrm{p}}$ and rBC mass $m_{\mathrm{rBC}}$, $\partial^2 N/\partial\log m_{\mathrm{p}}\partial\log m_{\mathrm{rBC}}$, from which the distributions of coating mass or coating mass ratio on each rBC particle can also be calculated ($\partial^2 N/\partial\log m_{\mathrm{coating}}\partial\log m_{\mathrm{rBC}}$). Details on the inversion scheme used to calculate the distribution from the 215    measurements are given in detail in Naseri et al. (2021a). In brief, the $m_{\mathrm{p}}$–$m_{\mathrm{rBC}}$ distribution is found by inverting a double convolution where the input data is the number concentration of rBC particles detected within each SP2 bin ($N_i$) at each CPMA set point $i$,

$$N_i = \int\limits_0^\infty \int\limits_0^\infty K\left(\mathbf{r}_i^*, m_{\mathrm{p}}, m_{\mathrm{rBC}}, d_{\mathrm{m}}\right) \frac{\partial^2 n}{\partial\log m_{\mathrm{p}}\partial\log m_{\mathrm{rBC}}} \mathrm{d}\log m_{\mathrm{rBC}}\mathrm{d}\log m_{\mathrm{p}} + \varepsilon_i, \tag{3}$$

where

$$K\left(\mathbf{r}_i*, m_{\mathrm{p}}, m_{\mathrm{rBC}} d_{\mathrm{m}}\right) = \sum_{\phi=1}^\infty f\left(\phi, d_{\mathrm{m}}\right) \Omega\left(m_{\mathrm{p},i}*, m_{\mathrm{p}}, d_{\mathrm{m}}, \phi\right) \Lambda\left(m_{\mathrm{rBC},i}*, m_{\mathrm{rBC}}\right) \tag{4}$$

is the kernel function, $\Omega$ is the CPMA transfer function, $\Lambda$ is the effective SP2 transfer function, $f\left(\phi, d_{\mathrm{m}}\right)$ is electric charge fraction for charge state $\phi$, $d_{\mathrm{m}}$ is the equivalent mobility diameter of the particles, $\mathbf{r}_i^* = \left[m_{\mathrm{p},i}^*, m_{\mathrm{rBC}i}^*\right]^\top$ is a vector holding the set points of the CPMA and the discretized basis set of the SP2 data, respectively; and $\varepsilon_i$ is the error in the measurement.

In this study, the model of Sipkens et al. (2020) (Case 1C) was used to calculate the CPMA transfer function and the 225    model of Wiedensohler (1988) was used for the charge fraction. The inversion used a solution resolution of 64 bins per decade, a Bayesian model for regularization parameter selection, and the exponential distance method for imposing distribution smoothness on the solution. Details on how other inversion methods affect the solution are found in Naseri et al. (2021a).

### 3.3    Comparison of CPMA–SP2 and SP2-only data

We analyzed the CPMA–SP2 and SP2-only results according to their traditional methods of presentation below. We also 230    converted between the two methods, to allow a direct comparison of differences in the two sets of results. To this end, a conversion from either rBC mass ($m_{\mathrm{rBC}}$) and coating mass ($m_{\mathrm{coating}}$) to rBC mass-equivalent diameter ($d_{\mathrm{rBC}}$) and coating thickness ($t_{\mathrm{coating}}$) or vice versa is required for comparison. The core-shell type morphology with core and shell material densities assumptions are required to convert CPMA–SP2 measurements (*i.e.,* $m_{\mathrm{p}}$ and $m_{\mathrm{rBC}}$) to coating thickness ($t_{\mathrm{coating}}$)





and rBC mass equivalent diameter ($d_{\text{rBC}}$). The $d_{\text{rBC}}$ is calculated by Equation 2, and $t_{\text{coating}}$ can be derived from

$$t_{\text{coating}} = \frac{1}{2}\left(\left(\frac{6\left(m_{\text{p}} - m_{\text{rBC}}\right)}{\pi\rho_{\text{coating}}}\right)^{\frac{1}{3}} - d_{\text{rBC}}\right), \tag{5}$$

where $\rho_{\text{coating}}$ is coating density and is assumed to be $1000 \text{ kg/m}^3$, to match the assumptions made in the LEO analysis (*c.f.,* Section 3.1.1).This assumption is evaluated later in Section 4.3. Note that, unlike the SP2–LEO method, the CPMA–SP2 method needs no optical model or assumed refractive indices to derive the coating thickness.

## 4   Results and Discussion

### 4.1   LEO detection limits

Figure 2 shows two-dimensional distributions of Case II measured by the CPMA–SP2 (green images; panels a and b) and LEO (blue images; panels c and d) methods. The distributions are presented both in terms of coating thickness versus rBC mass-equivalent diameter (Figure 2*a* and *c*) and in rBC mass versus total particle mass (Figure 2*b* and *d*). The rBC mass and total mass are both measured directly by the CPMA–SP2 method, while the coating thickness and mass equivalent diameter are typically shown in the LEO literature. As such, we have included both types of plots for comparison.

Figure 2*a* and 2*c* demonstrate the relationships between rBC diameter and coating thickness derived from the CPMA–SP2 and the LEO analysis, respectively. The detection limits of the LEO method are indicated by red lines in Figure 2, and are defined as follows. The light scattering detection (LSD) and broadband incandescence detection (BID) limits are defined as the lowest and highest amount of light that the optics and detectors can collect and will vary between SP2 units and models. For the SP2 used here (the original SP2 model), the limits are equivalent to about 150 nm to 430 nm in optical diameter, and 74 nm to 254 nm in rBC mass-equivalent diameter (0.38 to 15.36 fg), respectively. For comparison, the SP2 manufacturer currently reports detection ranges of 200 nm to 430 nm in optical diameter and 70 nm to 500 nm in rBC mass-equivalent diameter; *i.e.,* a much larger upper limit in rBC diameter (Droplet Measurement Technologies, 2023). Therefore, lines *iii* and *iv* in Figure 2 are simply the low and high BID limits of the SP2. Line *ii* is the upper coating detection limit of the LEO method and it is limited by the saturation of the scattering detector.

Since LEO is performed at a small fraction of the total scattering signal for a given particle, it increases the detection limit (*ii*) for the optical sizing of non-absorbing particles. The upper coating thickness limit of LEO (line *ii*), was $\sim 285$ nm in our study, due to saturation of the scattering detector. The lower coating-thickness limit of LEO (Line *i*) is due to a combination of both lower LSD and BID limits. Because coating thickness is defined from the sum of rBC diameter and optical diameter (Equation 1), the limit *i* means that the lowest coating thickness detectable increases with decreasing rBC diameter. This increase occurs because the additional light scattered by the rBC core raises the scattering signal above its detection limit. The exact boundary of the lower coating thickness limit of LEO is difficult to determine, especially because not only the LSD but also the split detector must be above the LOD. However, the endpoints of the limit are simply the lower coating thickness limit of LEO at the BID limits (Line *i*) (*i.e.,* lowest detectable coating thickness at low BID limit, and zero coating thickness at high





**Figure 2.** Distributions of rBC diameter and coating thickness (a, c), and total particle and rBC mass (b, d) of the same wildfire plume. Variants of the CPMA–SP2 measurements in panels (a) and (b) are shown in green. The variants of LEO analysis in panels (c) and (d) are shown in blue. Red lines in panels a and c indicate the detection limits, with *iii* and *iv* showing the low and high broadband incandescence detection limits, respectively, for both LEO and CPMA–SP2 methods, *i* and *ii* showing the LEO lower and higher coating detection limits. Grey shading indicates the physically impossible region where $m_\mathrm{p} > m_\mathrm{rBC}$.

BID limit) and is shown as a linear dashed line in the figure. The data clearly show that the true lower limit of coating-thickness quantification is much higher, since no measurements come near to Line *i* in Figure 2*c*.

To make a general comparison between the LEO results and those derived directly by the CPMA–SP2 measurements without making any morphological and density assumptions (Figure 2*b*), the distribution of Figure 2*c* was mapped onto $m_\mathrm{p}$—$m_\mathrm{rBC}$





mass space (Figure 2$d$) by rearranging Equation 5 to find $m_{\mathrm{p}}$. It is not physically possible to have a particle in which the rBC
mass exceeds the total mass of the particle; thus the reconstruction elements for which $m_{\mathrm{p}}$ is greater than $m_{\mathrm{rBC}}$ are prohibited
during the formal inversion and are greyed out in the figure.

A comparison of the particle mass range ($m_{\mathrm{p}}$) of distributions represented in Figure 2$b$ and $d$ measured by the CPMA–SP2
and the LEO measurements, respectively, demonstrates that the LEO reconstruction range is far more limited than that of the
CPMA–SP2 method. This deficiency is rooted in LEO's structural dependence on the scattering signal, which results in a high
LOD for $m_{\mathrm{p}}$, which can make using the LEO model inefficient when dealing with particles with low rBC mass and thin- to
moderate-coatings.

This point can be made clearer by a closer look at the mixing states of uncoated to heavily coated rBC particles from LEO
and CPMA–SP2 measurements. Figure 3 shows three examples of $m_{\mathrm{p}}$— $m_{\mathrm{rBC}}$ distributions, representing rBC-containing
particles with mostly thin to no coatings (Case I, Figure 3$a$ and $d$), as compared to a mixture of rBC particles with no coating
to moderate and heavy coatings (Case II, Figure 3$b$ and $e$), and rBC-containing particles mostly with heavy coatings (Case III,
Figure 3$c$ and $f$) [2]. In these plots, the main diagonal corresponds to the mass fraction of rBC of one, *i.e.,* $m_{\mathrm{rBC}}/m_{\mathrm{p}} = 1$, with
any line parallel to it corresponding to lines of constant rBC mass fraction of less than one, representing coated particles (lines
of rBC mass fractions of $m_{\mathrm{rBC}}/m_{\mathrm{p}}$ =0.75, 0.5, 0.25 and 0.05 are shown in Figure 3). Consequently, a part of distributions
that are clustered along the diagonal line represents rBC-containing particles with thinly to no coatings. Overall, Figure 3$a$-$c$
shows bimodal-bivariate distributions in which the distribution with a smaller particle mass mode corresponds to rBC particles
with thin coatings (small rBC to coating mass ratio; $m_{\mathrm{coating}}/m_{\mathrm{rBC}} \ll 1$), and the second mode represents rBC particles with
moderate to heavy coatings ($m_{\mathrm{coating}}/m_{\mathrm{rBC}} \gg 1$). As the relative concentration of rBC particles with thin and moderate-to-
heavy coatings varies mostly due to changes in wildfire conditions, moving from the left distributions to the right ones in
Figure 3, the second mode becomes wider and dominates the first mode.

It can be seen from Figure 3 that considerably more information on the mixing states of rBC-containing particles can be
measured from CPMA–SP2 measurements than from the LEO analyses, as LEO is biased towards the rBC with heavy coatings.
The reason for this disparity is that rBC particles with thin-to-moderate coatings may not be measured by the LEO method
because of the lower LEO detection limit (line $i$). Thus, LEO analysis may only describe a fraction of rBC-containing particle
population.

Apart from a more limited detection range of LEO, a comparison of distributions represented in Figure 2$a$ and $c$, show
that LEO data is concentrated near the higher LSD limits (*ii*) in Figure 2$c$; however, the CPMA–SP2 results suggest there are
no particles there. This indicates that the LEO method over-predicts the coating thickness of rBC-containing particles, which
stems from the assumptions made in LEO analysis, *e.g.,* the core-shell morphology. Besides, there are also some rBC particles
with a total particle mass of $\sim$ 4–20 fg that is clustered near the 1:1 line in each case, which is an artifact discussed in detail

---

[2]Figure A1 in Appendix provides an alternative viewpoint on the number distributions depicted in Figure 3, highlighting the mass concentration distributions for cases I, II, and III in relation to the total particle–rBC mass ($m_{\mathrm{p}}$—$m_{\mathrm{rBC}}$).





**Figure 3.** The distributions of number concentration for cases I, II, and III are depicted as functions of the total particle-rBC mass (denoted as $m_\mathrm{p}$—$m_\mathrm{rBC}$). These distributions have been measured through two distinct techniques: (a-c) the CPMA–SP2 method and (d-f) the LEO analysis. Main diagonal lines indicate pure rBC particles with a rBC mass fraction of one ($m_\mathrm{rBC}/m_\mathrm{p} = 1$), while parallel lines represent constant rBC mass fractions less than one, signifying coated particles ($m_\mathrm{rBC}/m_\mathrm{p} < 1$).

in Section 4.3. Additionally, a closer look at Figures 2 and 3 shows that the number concentration of the LEO analysis is much lower that the CPMA–SP2. These two issues are discussed in the following sections.





## 4.2 LEO counting statistics limitation

There are three common types of rBC-containing particles with reliable incandescence signals whose scattering signals escape quantification by LEO analysis:

1. Very small rBC-containing particles, with negligible scattering signals (below Line *i* in Figure 2). These result in noisy fits and unreliable sizing.

   2. Very large rBC-containing particles, with very large scattering signals (above Line *ii* in Figure 2) which saturate the LSD.

   3. rBC particles that evaporate before their position in the laser beam can be detected by the split detector.

The effect of excluding these rBC-containing particles from the LEO analysis can be seen in Figure 4, where the size distribution of rBC-containing particles having valid scattering data and are measurable by the LEO is compared with the one directly measured by the incandescence signal of the SP2. Specifically, Figure 4 shows the number concentration distributions of particle mass ($\mathrm{d}N/\mathrm{dlog}m_\mathrm{p}$) and rBC mass ($\mathrm{d}N/\mathrm{dlog}m_\mathrm{rBC}$), respectively, for the LEO and CPMA–SP2 methods which are found by integrating the two-dimensional distributions of Figure 2*b* and *d* over $m_\mathrm{p}$,

$$\frac{\mathrm{d}N}{\mathrm{dlog}m_\mathrm{rBC}} = \int\limits_0^\infty \frac{\partial^2 N}{\partial \log m_\mathrm{p}\partial \log m_\mathrm{rBC}} \mathrm{dlog}m_\mathrm{p}, \tag{6}$$

and by integrating over $m_\mathrm{rBC}$,

$$\frac{\mathrm{d}N}{\mathrm{dlog}m_\mathrm{p}} = \int\limits_0^\infty \frac{\partial^2 N}{\partial \log m_\mathrm{p}\partial \log m_\mathrm{rBC}} \mathrm{dlog}m_\mathrm{rBC}. \tag{7}$$

Figure 4 therefore shows the overall rBC-containing particle size distribution in two different ways, first as a function of total particle mass (Figure 4a) and second as a function of the rBC mass within the particle (Figure 4b). Both Figure 4a and Figure 4b
show that the overall number concentration (area under the curves) of rBC-containing particles detected by LEO analysis is only $\sim 50\%$ of that directly measured from the incandescence signal of the SP2, because of the higher LOD of the scattering signal. Figure 4c and d show the particle mass and rBC mass distributions (*i.e.,* $\mathrm{d}M_\mathrm{p}/\mathrm{dlog}m_\mathrm{p}$ , and $\mathrm{d}M_\mathrm{rBC}/\mathrm{dlog}m_\mathrm{rBC}$) of the same data and they show that LEO only measures $35\%$ of the total particle mass concentration ($M_\mathrm{p}$).The data presented also reveals that the median values of total particle mass ($m_\mathrm{p}$) and refractory black carbon mass ($m_\mathrm{rBC}$) obtained through the SP2-
LEO method are approximately half those obtained through the CPMA-SP2 method. This observation suggests a pronounced bias in the distribution of particle masses towards lower values when utilizing the SP2-LEO method.

   In Figure 4a, the size distribution of LEO-analyzed particles drops to zero at a smaller total particle mass than does the size distribution of all particles, because of detector saturation. In contrast, Figure 4b shows the LEO-analyzed particle size





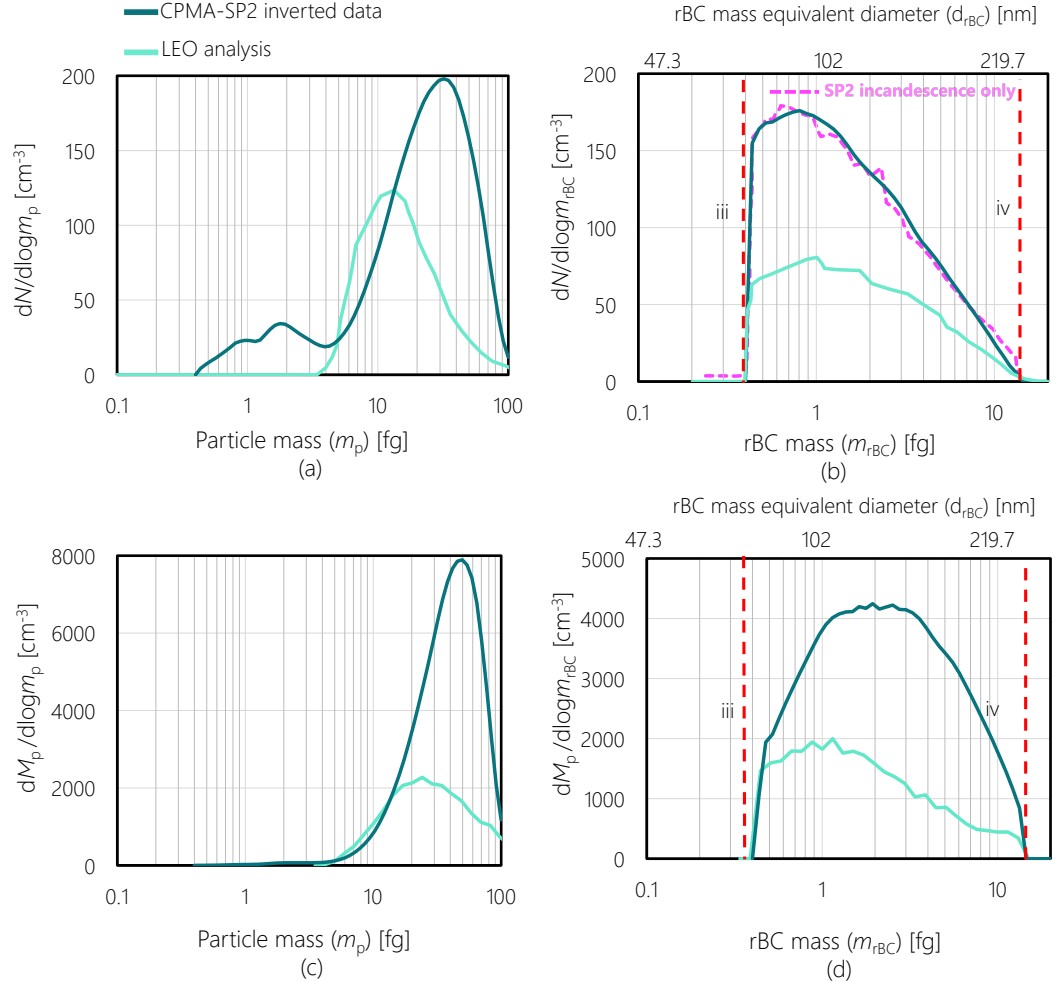

**Figure 4.** Comparison of rBC-containing particle number and mass concentrations in Case II are illustrated in panels (a) and (c) as functions of total particle mass, and in panels (b) and (d) as functions of rBC particle mass. These measurements were acquired using both the LEO analysis and the CPMA-SP2 method. The boundaries of low and high BID limits are indicated by red dashed lines of *iii* and *iv*, respectively. The purple dashed line represents the number concentration of rBC particles directly derived from the incandescence signal of the SP2.

distribution with a near-constant offset in amplitude from the size distribution of all particles, because SP2–LEO excluded very large and very thin coatings from the data set. We note that the CPMA–SP2 data inversion matches the SP2-total size distribution almost perfectly because of constraints placed on the inversion of the CPMA–SP2 measurements. Overall, we conclude that LEO method only provides mixing state information on a small and biased subset of the entire rBC population.





### 4.3 Evaluation of the accuracy of LEO coating thickness measurement

A comparison of the particle mass ($m_\mathrm{p}$) distributions illustrated in Figure 2 to 4 shows that the LEO particle mass modes are

noticeably larger than the corresponding CPMA–SP2 ones. One way to evaluate the accuracy of LEO coating-thickness calculations is to measure coated rBC particles of known mass. We did this by using the CPMA to select particles (known particle mass) and conducting LEO analysis on the CPMA-classified particles. In such an experiment, the particle mass determined by LEO analysis should fall within the mass range given by the CPMA set-point ($m^*$) (*i.e.*, $m_\mathrm{p} \in qm^* \times (1 \pm 1/R_\mathrm{m})$; where $R_\mathrm{m}$ is the CPMA resolution).

We, therefore, performed LEO analysis on CPMA-classified particles at a mass set point of $m^* = 14.38$ fg with $R_\mathrm{m}$=9 (we chose this set point because it corresponded to a large number of classified particles downstream) for Case I and Case III. The results are shown in Figure 5. Figure 5*a* and 5*b* shows the normalized $m_\mathrm{p}$— $m_\mathrm{rBC}$ distributions of CPMA-classified particles, along with their corresponding marginal distributions (*i.e.*, $\mathrm{d}N/\mathrm{d}\log m_\mathrm{p}$, Equation 7), that were determined by LEO for rBC particles in Case I (where particles displayed thin or negligible coatings) and Case III (where particles displayed

moderate-to-heavy coatings) using a coating refractive index of $n_\mathrm{coating} = 1.5$ and $\rho_\mathrm{coating}$=1000 kg/m$^3$.

If the rBC-containing particles were homogeneous in shape and physico-chemical composition, the distribution of CPMA-classified particles in Figure 5 would exhibit two modes, representing singly- and doubly-charged particles [3]. These two modes would appear as vertical lines, and their predicted location is shown using red dashed (singly-charged) and dotted (doubly-charged) lines in the figure. These modes were predicted using reasonable assumptions of density and refractive index, which

are discussed further below.

Figure 5 clearly shows that only a subset of particles, at low rBC mass ($m_\mathrm{rBC} < 2$ fg), falls close to the predicted vertical modes. At moderate masses ($m_\mathrm{rBC}$ between 2 and 10 fg), the LEO-calculated particle mass consistently falls below the predicted mass. At higher masses ($m_\mathrm{rBC} > 10$ fg), the LEO-calculated mass returns to the predicted mass. This trend is evident for singly-charged particles in all panels of Figure 5, and is also evident for the doubly-charged particles in Figure 5b, due to its

higher signal-to-noise ratio, resulting from higher number concentrations. This trend can be described as an "S" shape in the normalized $m_\mathrm{p}$— $m_\mathrm{rBC}$ distributions, and is illustrated with a black outline.

The "S" shape in Figure 5 indicates significant and systematic variations in particle masses determined by LEO analysis. The potential sources of these variations may be attributed to different factors:

- i. Assumptions embedded within the theoretical model: The LEO analysis assumes that the optical properties of coated
rBC can be accurately described using a core-shell Mie-theory model. Therefore, the accuracy of the LEO results are limited by the accuracy of the core-shell Mie model.

- ii. Assumptions about the physical properties of coatings: The accuracy of LEO analysis relies on assumptions made regarding the physical properties of coatings, such as their density and refractive index. Any discrepancies between the assumed properties and the actual properties of the coatings present on the particles could lead to inaccurate mass deter-

---

[3]Triply-charged particles are also possible, but are omitted from this discussion for simplicity.



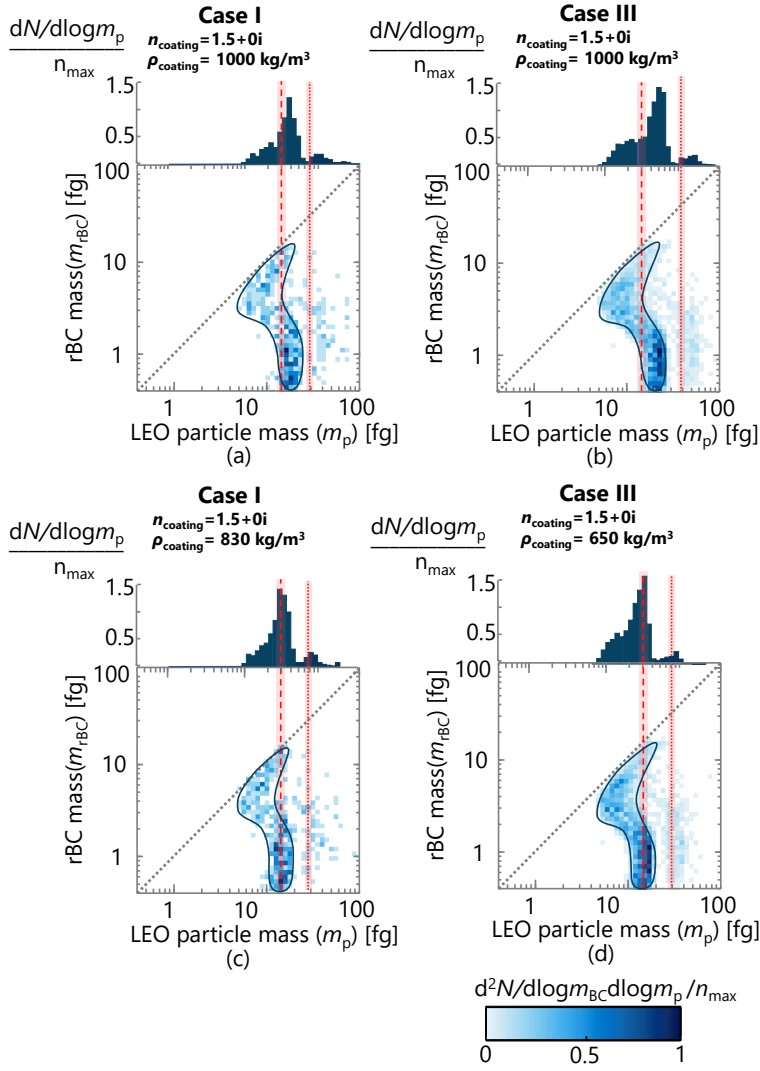

**Figure 5.** The normalized $m_\mathrm{p}$— $m_\mathrm{rBC}$ distributions of mass-classified particles at $m^* = 14.38$ fg that were analyzed by LEO for rBC-containing particles with (a,c) thin or no coatings (Case I) and (b,d) moderate-to-heavy coatings (Case III) using the coating refractive index of $n_\mathrm{coating} = 1.5$ and (a,b) $\rho_\mathrm{coating}$=1000 kg/m$^3$ or the coating effective densities of $\rho_\mathrm{eff,c}$=830 and 650 kg/m$^3$ derived for (c) Case I and (d) Case III at the given coating refractive index, respectively.





minations. These models rely on certain assumptions about particle structure and composition, and deviations from the actual particle configurations can introduce errors in mass determination.

In the following, we discuss these two factors in detail using additional calculations.

### 4.3.1 Accuracy variations due to the core-shell model

Accuracy variations due to the core-shell model, which might lead to the "S" shaped variation in the LEO-calculations shown
in Figure 5$a$ and $b$, may be further visualized by plotting the ratio of $m_{\mathrm{LEO}}$ and $m_{\mathrm{p}}$ (which would be 1.0 for perfect LEO calculations) as a function of the coating mass to rBC mass ratio, as shown in Figure 6. $m_{\mathrm{LEO}}$ is the total particle mass implied by the SP2–LEO analysis:

$$m_{\mathrm{LEO}} = \frac{\rho_{\mathrm{coating}}\pi}{6}\left(\left(d_{\mathrm{rBC}} + 2t_{\mathrm{coating}}\left(\mathrm{RI}\right)\right)^3 - d_{\mathrm{rBC}}^3\right)$$
$$= \frac{\rho_{\mathrm{coating}}\pi}{6}\left(d_{\mathrm{opt}}{}^3(\mathrm{RI}) - d_{\mathrm{rBC}}{}^3\right) \tag{8}$$

We chose to plot $m_{\mathrm{LEO}}$ against $m_{\mathrm{p}}$ because they are directly analogous to the calculations of Liu et al. (2017), who also used a
CPMA to measure $m_{\mathrm{p}}$ directly. Those authors reported their results in terms of the ratio of calculated-to-measured scattering cross sections, rather than the ratio of calculated-to-measured particle mass, and reported on a slightly different range of particle sizes. Here, we chose to retain mass units as this is the focus of our work.

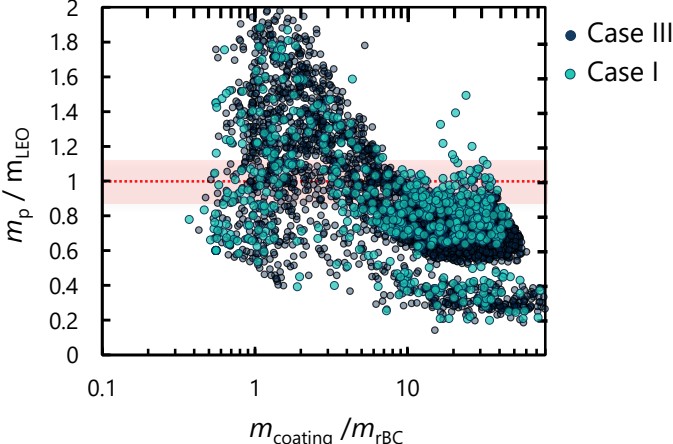

**Figure 6.** Particle mass ($m_{\mathrm{p}} \approx 14.38 \pm 0.11$) expected downstream of the CPMA relative to the particle mass calculated by the SP2–LEO method ($m_{\mathrm{LEO}}$) for various coating mass ratios ($m_{\mathrm{coating}} = m_{\mathrm{p}} - m_{\mathrm{rBC}}$). The increase in $m_{\mathrm{p}}/m_{\mathrm{LEO}}$ for small coatings indicates undersizing by SP2–LEO.

Liu et al. (2017) found that the core-shell model had the highest accuracy (*i.e.*, the best match to the measured scattering cross-sections) for uncoated BC particles with mass ratio below 0.1 or greater than 3.0, for particles of total mass 2 fg. Liu
et al. (2017) also pre-selected particles with the CPMA at 1–10 fg and obtained similar results. In our study, as we pre-selected




particles with a total mass of $m_\mathrm{p} = 14.38\,\mathrm{fg}$, and these mass ratio values correspond to mass ratios of $m_\mathrm{rBC}/m_\mathrm{p} > 0.91$ and $m_\mathrm{rBC}/m_\mathrm{p} < 0.25$, respectively, and to $m_\mathrm{rBC} > 13.1\,\mathrm{fg}$ and $m_\mathrm{rBC} < 3.6\,\mathrm{fg}$, respectively. In between this range, their accuracy followed an "S" shape similar to our Figure 5; the core-shell model first overpredicted and then underpredicted scattering. Liu et al. (2017) showed that no simple optical model could capture this trend, and noted that the assumed material density of the coating affected the results.

We note that the trends observed in Figure 6 and similarly by Liu et al. (2017) are consistent with soot-particle compaction due to thin coatings. If scattering by the coatings are minor, a decrease in light scattering cross-section upon compaction is expected (Liu and Mishchenko, 2018) at the 45° and 135° scattering angles measured by the SP2 (Gao et al., 2007). This would lead to an undersizing by SP2–LEO and an increase in the $m_\mathrm{p}/m_\mathrm{LEO}$ ratio, as observed. Later, as coating mass ratios increase to 5 or higher, for which spherical particles have been observed (*i.e.,* at which rBC is fully encapsulated) (Corbin et al., 2023, and citations therein), the undersizing by SP2–LEO is reduced. Some spread in the data is expected according to this phenomenon, as the size change of soot with compaction is larger for larger soot particles.

### 4.3.2   Accuracy variations due to varying particle properties

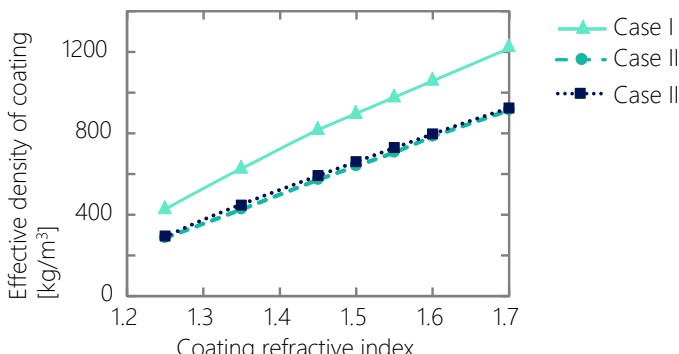

**Figure 7.** Pairs of effective coating densities and refractive index of coating materials used in Mie calculations that match the particle mass modes of various rBC-containing particles retrieved by the LEO method at a CPMA mass set point of 14.38 fg.

It is possible that the inaccuracy of the LEO-calculated particle mass in Figure 5 corresponds to a variation in particle physical properties. To explore this possibility, we solved Equation 5 for the coating density that would be required for the LEO-calculated mode of particle mass to match the CPMA mass set point. For simplicity, we focus here on particles with rBC mass $m_\mathrm{rBC} < 1\,\mathrm{fg}$.

Figure 5*c* and *d* show one example of this calculation. The figures show the adjusted $m_\mathrm{p}$— $m_\mathrm{rBC}$ LEO distributions of CPMA-classified particles using effective coating densities at a fixed coating refractive index of $n_\mathrm{coating} = 1.5$. As shown in Figure 5*c* and *d*, an effective coating density of 830 kg/m³ was needed to adjust the peak of small singly-charged particles with



$m_\mathrm{rBC} < 1\,\mathrm{fg}$ to the correct particle mass for Case I. In contrast, an effective coating density of 650 kg/m³ was needed for Case III.

For large particles with $m_\mathrm{rBC} > 1\,\mathrm{fg}$, these effective coating densities clearly overcompensate for the LEO inaccuracy. The mass of these particles was underpredicted at 1000 kg/m³, and is even further underpredicted at the effective densities obtained for small particles (*i.e.,* 650 kg/m³ and 830 kg/m³ for Case I and III).

In Figure 7, the refractive-index–effective-density pairs which would align the LEO-calculated mass (at $m_\mathrm{rBC} < 1\,\mathrm{fg}$) with the CPMA-classified mass are shown. Since this calculation allowed for two free parameters, any value along the plotted curves would be a valid result. However, no single pair of values could be used for the three cases, implying a significant difference in physical or morphological properties between them.

The physical cause behind the difference in physical or morphological properties in Case I compared to Cases II and III is beyond the scope of this work, but we will briefly mention some possibilities. Many factors, like the type of biomass and its moisture content, and any influence on the pre-ignition pyrolysis phase would change the physical and chemical composition of the particles, and, consequently, the material density of organics emitted from the combustion (which may vary by e.g., ∼30 %, Hu et al., 2021). Also, the particle concentrations in Case I are substantially lower that the other cases, which likely results in a higher fraction of rBC particles originating from Diesel engines rather than wildfire smoke, which likely have different physical and optical properties. It is also seen that for Case II and III, which are primarily composed of thickly coated rBC particles presumably emitted from wildfire smoke, the coating densities are quite low, ranging from 350 to 900 kg/m³ over a range of coating refractive index of 1.25 to 1.7. However, studies have shown that the material density of organics in wildfire smoke ranges from 800 to 1600 kg/m³ (Turpin and Lim, 2001; Nakao et al., 2013; Li et al., 2016; Hu et al., 2021). These densities are as low as the effective densities measured for soot agglomerates with highly open structures, which are extremely unlikely values for coatings.

Our analysis implies that the range of effective densities shown here are not the material density of the coating, but rather an effective density that reconciles the errors in the SP2–LEO analysis. Specifically, low effective densities would be required to balance an overprediction of particle volume (diameter) by SP2–LEO. This overprediction may be due to the core-shell sphere assumption inherent in LEO, which is known to lead to overpredicted scattering relative to models of core-shell coated aggregates (Wu et al., 2014). On the other hand, attached or partially-embedded morphologies lead to slightly less light scattering and do not explain our observed trends (Liu and Mishchenko, 2018).

Another possibility is that these particles are the 1064-nm-absorbing tarballs discussed in previous SP2 literature (Corbin et al., 2019a; Corbin and Gysel-Beer, 2019). However, the effective density of such tarballs is not expected to be unusually low. So the apparent oversizing by SP2–LEO would require that tarballs swell substantially in the SP2 laser. Although swelling has been proposed to occur for soot during laser heating (Michelsen et al., 2015), Corbin and Gysel-Beer (2019) reported optical diameters of about 250 nm for tarballs measured with an SP2, which rules out the hypothesis that tarball swelling led to the SP2–LEO oversizing discussed here.

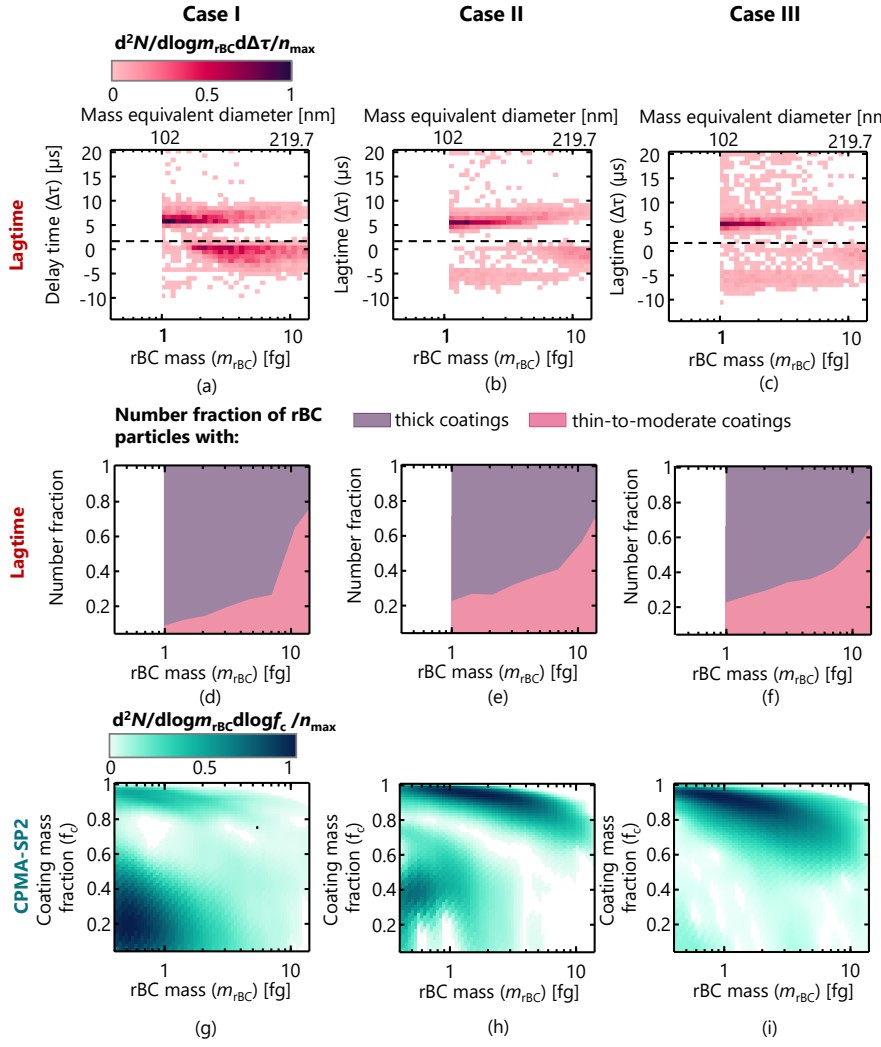

**Figure 8.** Normalized number concentration distributions of lag-times as a function of rBC mass for (a) Case I, (b) Case II and (c) Case III. The time-lag threshold of $\Delta\tau = 2\ \mu s$ is shown as a dashed line in panels (a) to (c). (d-f) The average proportions of rBC particles that have thick coatings and those with thin-to-moderate coatings. These proportions were determined using SP2 lag-time analysis. (g-i) The distribution of coating mass fraction ($f_c = m_{coating}/m_p$) as a function of rBC mass of particle populations measured by the CPMA–SP2 measurement system.





### 4.4 Lag-time limitations in coating estimation

Lag-time analysis of SP2 data is used qualitatively to classify rBC particles as thickly- or thinly-coated. The range and accuracy of the lag-time classification can be assessed by comparing it to CPMA–SP2 data. Figure 8a-c shows the normalized number concentration distributions of lag-times as a function of rBC mass for the three sample cases. The red shading represents the number fraction of particles falling within a histogram bin of rBC mass range and lag-time ($\mathrm{d}\log m_{\mathrm{rBC}}$ and $\mathrm{d}\Delta\tau$). As shown in Figure 8a-c the measured particles exhibit a distribution of lag-times, spanning negative and positive values, and

with multiple modes (most clearly seen in Case I). Figure 8d-f illustrates the average number fraction of particles with *thick* coatings (lag-time >2 μs) versus *thin-to-moderate* coatings (lag-time <2 μs) for an rBC mass range $m_{\mathrm{rBC}} \sim 1$ fg to 15.4 fg (rBC mass-equivalent diameter $d_{\mathrm{rBC}} \sim$102 nm—254 nm). The results shown in Figure 8d-f indicate a higher number fraction of particles exhibiting *thick coatings* for smaller rBC masses, and this fraction decreases as the rBC masses increase. The number fraction of particles in each category remains relatively consistent across all three cases.

Figure 8g-i, on the other hand, shows the normalized number distributions of coating mass fraction ($f_{\mathrm{c}} = m_{\mathrm{coating}}/m_{\mathrm{p}}$) as a function of rBC mass for all three cases measured by the CPMA–SP2 system. These distributions are derived by transforming the $m_{\mathrm{p}}$—$m_{\mathrm{rBC}}$ distributions of Figure 3 into coating mass fraction — rBC mass space.

Comparing the CPMA–SP2 results in panel Figure 8g-i with the corresponding lag-time results presented in Figure 8d-f, it becomes clear that the lag-time analysis provides only an extremely rough estimate of coating thicknesses. For example, while

lag-time analysis indicated that thick coatings prevailed in Case I, the CPMA–SP2 data make it clear that most of the small particles were thinly coated. Since lag-time analysis is limited to the largest particles, there is a severe bias in the estimated mean coating-thickness number fraction. This leads to the conclusion that the lag-time analysis should be used with great care, as it does not have the capability to quantitatively estimate coating thicknesses for the vast majority of rBC particles.

### 4.5 Temporal resolution of SP2 measurement methods

Figure 9 shows the differences between the $m_{\mathrm{p}}$—$m_{\mathrm{rBC}}$ distribution of smoke particulates of Case III derived from a ~80-minute CPMA–SP2 measurement (Figure 9a) and that obtained by ~ 1-minute SP2 LEO measurements performed (Figure 9b) before (t= 0 min), (Figure 9c) during (t= 30 min; between the CPMA set points of 5.2 fg and 8.6 fg), and (Figure 9d) after the CPMA–SP2 scan (t= 80 min). The CPMA–SP2 measurement was performed in such a way that the CPMA mass set point was scanned in an ascending order (*i.e.,* CPMA was stepped from the low CPMA mass set point of 0.4 fg to the high

mass set point of 100 fg over ~80 minutes). The $m_{\mathrm{p}}$—$m_{\mathrm{rBC}}$ distribution in Figure 9a is normalized by the total number concentration of rBC-containing particles derived from the CPMA–SP2 measurement, whereas the $m_{\mathrm{p}}$—$m_{\mathrm{rBC}}$ distributions in Figure 9b-d are normalized by the maximum total number concentration of particles found in the LEO measurements. The number concentration of rBC-containing particles increased by ~ 3.7 times over the first 30 min and ~5.6 times over the entire time of the CPMA–SP2 scan. The CPMA–SP2 inversion assumes that the number concentration of particles is constant with

respect to time over the course of the CPMA–SP2 scan. As is the case here, this assumption is not always true. Since the CPMA





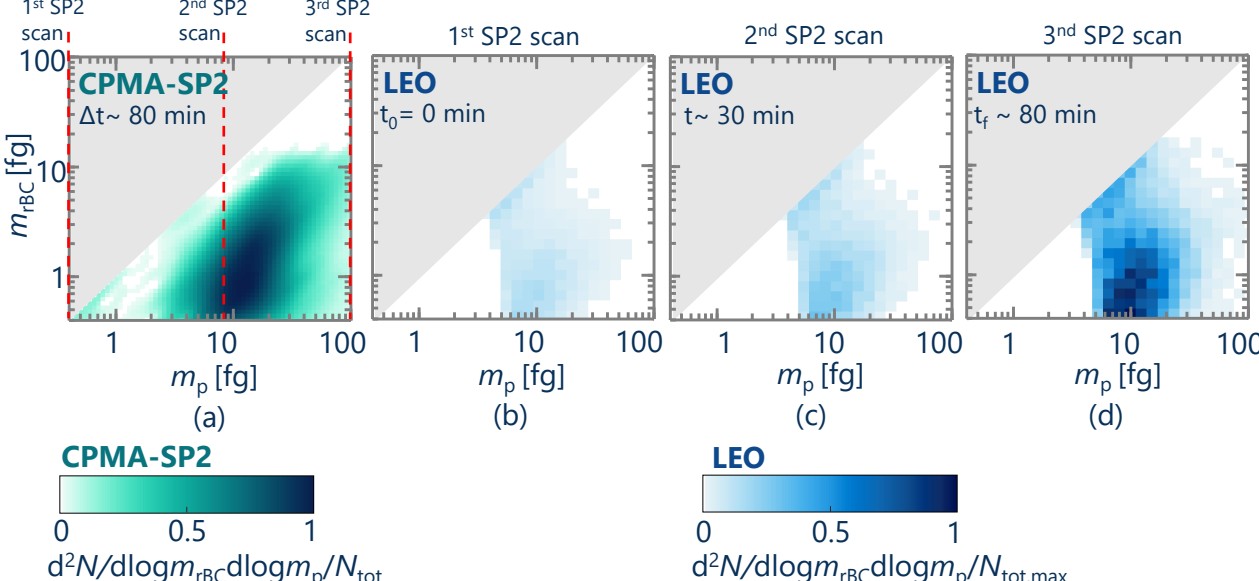

**Figure 9.** Comparison of the average $m_p$—$m_{rBC}$ distribution of smoke particulates of Case III derived from (a) CPMA–SP2 measurement and the average distribution obtained by instantaneous LEO measurements performed (b) before, (c) during, and (d) after the CPMA–SP2 scan.

set points were stepped upward in mass, the number concentration of low-mass particles is underestimated, while the number concentration of high-mass particles is overestimated relative to time-averaged LEO measurements.

The compromise of the CPMA–SP2 approach is that it is slower, typically requiring over 30 minutes per scan, compared to a few minutes for SP2-only measurements (both durations depend on the number concentration of particles and guidelines for optimal scan times for CPMA-SP2 are given by Naseri et al. (2021b)). Therefore, for measurements of samples that are highly transient, as in the case of high-altitude research aircraft the CPMA–SP2 method would require a sample reservoir.

## 5 Conclusions

The present study investigated the performance of various SP2 measurement methods in measuring rBC mixing states that are primarily caused by wildfire smoke. This was accomplished using example data with a broad range of rBC mixing states. The rBC mixing states were characterized with a particular focus on the tandem CPMA–SP2 measurements and SP2-only measurements using LEO analysis that map out the rBC-containing particle characteristics in two-dimensional mass space. Comparison of the results of the LEO analysis and the CPMA–SP2 indicates that the LEO measurement range is very limited, because it relies on the less-sensitive light-scattering signal as well as the position-sensitive light-scattering detector. These limitations, along with the assumptions on which LEO is based (*e.g.,* core-shell morphology and assumed density and refractive index for coating materials), bring about biased results and make the LEO method a qualitative measure, for a narrow range of



the rBC population. Additionally, the accuracy of the LEO and CPMA–SP2 methods in characterizing particles was analyzed. The evaluation of the accuracy of LEO coating thickness measurement reveals notable variations in particle masses determined by LEO analysis compared to CPMA-based measurements.

These discrepancies can be attributed to assumptions made about the physical properties of coatings and the theoretical models used in the LEO analysis for presenting the scattering behavior of BC particles. It is demonstrated that the core-shell model used in LEO analysis may not accurately represent the mixing arrangements of BC-containing particles, leading to errors in mass determinations. Efforts to adjust the LEO results by incorporating effective coating densities and refractive indices partially correct the measurements but still leave certain particles at physically-impossible masses. The possibility of different particle types and physico–optical properties further contributes to the challenges in accurately determining coating thickness using LEO analysis. As with other BC mixing states, wildfire smoke may exhibit non-uniform characteristics, it is not possible to identify a single physical and optical property for the LEO that can effectively address these limitations.

On the other hand, the CPMA–SP2 approach provides precise measurements of total particle mass. Therefore, it can be inferred that the CPMA–SP2 method demonstrates superior accuracy compared to the SP2-only method as it addresses and eliminates one of the limitations of the LEO technique, resulting in improved particle categorization. The compromise of the CPMA–SP2 method is that it is not a real-time measurement, typically requiring more than 30 minutes per scan for atmospheric measurements. Rapidly changing aerosol samples should therefore be captured in a reservoir prior to CPMA–SP2 measurement, to ensure valid results.

Finally, as a result of the high level of uncertainty in classifying rBC particles with invalid scattering signals, the lag-time analysis could not distinguish between the relative number fraction of rBC particles with thick and thin-to-moderate coating in the example data presented.

## Appendix A: Appendix

Figure A1 offers an alternative perspective on the number distributions found in Figure 3. It illustrates mass concentration distributions for cases I, II, and III, with a focus on their relationship to total particle-rBC mass ($m_{\mathrm{p}}$—$m_{\mathrm{rBC}}$). These distributions were determined through two distinct measurement techniques: the CPMA–SP2 method (panels a-c) and the LEO analysis (panels d-f).

*Author contributions.* A.N. formulated the study, conducted the experiments, performed comprehensive data analysis, and composed the initial draft of the paper. J.S.O and J.C.C. played an active role in result discussions, significantly contributed to data interpretation, and engaged in collaborative writing.

*Competing interests.* The authors declare no conflicts of interest relevant to this research or paper.



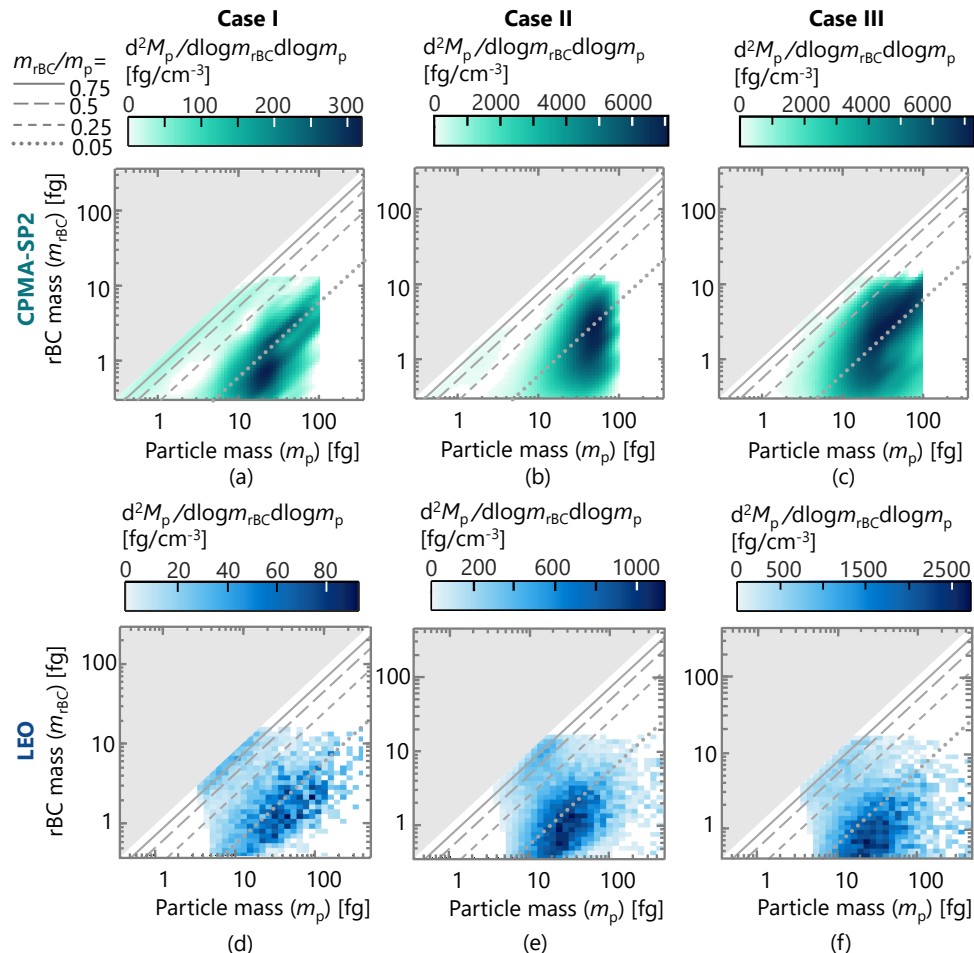

**Figure A1.** The distributions of mass concentration for cases I, II, and III are depicted as functions of the total particle-rBC mass. These distributions have been assessed through two distinct measurement techniques: (a-c) the CPMA–SP2 method and (d-f) the LEO analysis. Main diagonal lines indicate pure rBC particles with a rBC mass fraction of one ($m_{\mathrm{rBC}}/m_{\mathrm{p}} = 1$), while parallel lines represent constant rBC mass fractions less than one, signifying coated particles ($m_{\mathrm{rBC}}/m_{\mathrm{p}} < 1$).



510 *Acknowledgements.* The authors gratefully acknowledge funding from Alberta Innovates and NSERC (FlareNet Research Network), and support from the British Columbia Ministry of Environment and Climate Change as well as Transport Canada. More specifically, we would like to thank Steve Josefowich, without whose kind support and cooperation, this study would have been impossible.



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
