# Peer review of "Comparison of the LEO and CPMA-SP2 techniques for black-carbon mixing-state measurements"

_EGUsphere, 2023_

## Author Comment (AC1)

**Comparison of the LEO and CPMA-SP2 techniques for black-carbon mixing-state measurements**
**(Response to Reviewers)**

December 21, 2023

**Dear Reviewer,**

Thank you for providing us with your valuable feedback on our manuscript titled ***Comparison of the LEO and CPMA-SP2 techniques for black-carbon mixing-state measurements***, submitted to Atmospheric Measurement Techniques. We sincerely appreciate the time and effort you have dedicated to reviewing our work, and we are grateful for the opportunity to incorporate your constructive feedback into our revisions. After carefully considering all of your comments, we have made revisions throughout the manuscript to address them. To clearly indicate our responses to the reviewer's comments, we have highlighted them in red text text, while changes made to the manuscript are highlighted in blue text. We believe that these revisions significantly improve the quality and clarity of our manuscript. We are thankful for your valuable input, which has helped us enhance this work. We hope that our responses and revisions adequately address your concerns, and we eagerly await your positive consideration of our revised manuscript.

**Comment:** The manuscript gave the comparison results of the measured black carbon mixing states from LEO and CPMA-SP2 techniques. The articles fall well into the scope of AMT. In general, the results are interesting and important to the related researchers. The manuscript can be accepted after some major revisions.

**Major Comments**:

1) The LEO detection limits results and the corresponding impacts of measured black carbon mixing states were mentioned before by Zhao, Shen et al. (2020). The new findings of this manuscript relative to that article should be detailed.

   **Authors Response:** We acknowledge the reviewer's observation and appreciate the opportunity to clarify the relationship between our study and the work conducted by Zhao al. (2020). In response to this comment, we have revised our manuscript to explicitly detail the Zhao al.'s (2020) finding and compare it with the present study. The comparison between our findings and those presented by Zhao et al. (2020) is now elucidated in the revised version of our manuscript. Specifically, please refer to page 2, where the relevant content can be found in blue color between lines 55 and 60.

   Second, the range of response of the SP2 detectors does not span all relevant scenarios. For example, very small (but significant in mass concentration) rBC particles do not scatter enough light to be detected, so the otherwise broad range of response of the SP2 incandescence detector cannot be fully exploited (Taylor et al., 2015; Zhao et al., 2020).

2) The Gassion fit of LEO fits of the LSD signals may still be useful even though the signals detected by the SP2 are saturated. If these particles are considered, the difference between the LEO and CPMA-SP2 methods would not be so different as shown in Figure 4.

   **Authors Response:** We appreciate the reviewer's thoughtful comment. We would like to clarify that we indeed included all possible LSD signals in our analysis, We only excluded particles with scattering signals that saturated almost instantly, preventing the determination of the 3% threshold of the maximum laser intensity. In section 3.1 of the paper, we stated this as, "We defined the leading edge of the scattering signal as 3% of the maximum laser intensity based on scatter plots of LEO and standard analysis for non-absorbing particles."; therefore, we excluded signals which saturated at 3% of peak laser power, not signals which saturated at the peak laser power. To make this point clearer in the manuscript, bullet point 2 in section 4.2 (page 13), where the relevant content can be

found is modified as:

Very large rBC−containing particles, with very large scattering signals (above Line *ii* in Figure 2), saturate the LSD even before the 3% threshold of the maximum laser intensity is reached.

3) Around line 360, iff the BC core were not compacted, ie the BC core is filled with some air, then the effective density of the BC core may be reconsidered as shown in (Zhang, Zhang et al. 2016, Zhang, Su et al. 2018). This effects should also be dissucssed.
**Authors Response:** We acknowledge the reviewer's comment. Indeed, the detailed work of Zhang and colleagues has caught our attention. However, we avoided a detailed discussion of it because of the complexity of understanding the mobility diameters in relation to particle mass. Nevertheless, it should indeed have been cited. We now added the following statement to page 18 between lines 392 and 394.

Our conclusions here are also consistent with work relating soot-core morphology to coating thickness based on mobility-diameter based effective densities, which exploit particle mobility diameters to estimate particle volume in relation to SP2 scattering measurements Zhang et al. (2016, 2018b).
**minor comments:**

4) Line 209, it should be mentioned that some lag time of the detailed information can be found in Figure 8 or some following discussion
**Authors Response:** Noted and the below sentence is added to the end of the section 3.1 .
The detailed information pertaining to lag time can be found in Section 4.4, illustrated in Figure 8.

We believe that these revisions have improved the clarity of the manuscript. We hope that our responses and revisions adequately address your concerns and that you find the revised manuscript acceptable for publication in Journal of Atmospheric Measurement Techniques.

Thank you once again for your valuable feedback, which has helped us enhance the quality of our work. We look forward to your positive consideration of our revised manuscript.

---

## Author Comment (AC2)

**Comparison of the LEO and CPMA-SP2 techniques for black-carbon mixing-state measurements**
**(Response to Reviewers)**

December 21, 2023

**Dear Reviewer,**

Thank you for providing us with your valuable feedback on our manuscript titled ***Comparison of the LEO and CPMA-SP2 techniques for black-carbon mixing-state measurements***, submitted to Atmospheric Measurement Techniques. We sincerely appreciate the time and effort you have dedicated to reviewing our work, and we are grateful for the opportunity to incorporate your constructive feedback into our revisions. After carefully considering all of your comments, we have made revisions throughout the manuscript to address them. To clearly indicate our responses to the reviewer's comments, we have highlighted them in red text text, while changes made to the manuscript are highlighted in blue text. We believe that these revisions significantly improve the quality and clarity of our manuscript. We are thankful for your valuable input, which has helped us enhance this work. We hope that our responses and revisions adequately address your concerns, and we eagerly await your positive consideration of our revised manuscript.

**Comment:** Naseri et al. compare several techniques to derive the mixing state from black carbon measurements obtained by the single particle soot photometer (SP2). Their results show that coupling the SP2 with a centrifugal particle mass analyzer provides the best results. Overall the paper is well written and clearly explains the origin and application of the different methods. This manuscript presents a valuable addition to groups that seek to experimentally determine the mixing state of black carbon using similar techniques. I therefore recommend publication.

**Minor Comments**:

1) "normalized derivative of the scattering signal is used in the ND approach to obtain analogous information" -> Please explain what the normalized derivative is here.

   **Authors Response:** We acknowledge the reviewer's comment. In response to this comment, we added a brief description of the ND method to the beginning of section 3.1.2 :

   The normalized derivative (ND) approach, as introduced by Moteki and Kondo (2008), offers a methodology for assessing the time-dependent solid angle scattering cross-section ($\Delta C_{\mathrm{sca}}(t)$) to identify individual particles traversing through a Gaussian SP2 laser beam. This technique hinges on the concept that the normalized derivative of the scattering signal ($S'/S$), as detected by the SP2, can be broken down into the normalized derivatives of the incident irradiance ($I'/I$) and the scattering cross-section ($\Delta C'_{\mathrm{sca}}/\Delta C_{\mathrm{sca}}$). In the realm of evaporative particles, the equality $S'/S = I'/I$ holds true until evaporation initiates at a specific point within the laser beam. The incident irradiance $I(t)$ for individual particles is deduced from $I'/I$, extracted from $S'/S$. Consequently, $\Delta C_{\mathrm{sca}}(t)$ is derived from $I(t)$ and $S(t)$. The ND approach to evaluating rBC mixing states is similar to the LEO approach in that it extrapolates the initial particle size from the first portion of the scattering signal, except a different methodology is used to estimate the undisturbed particle diameter.

2) "The upper coating thickness limit of LEO (line ii), was $\sim 285$ nm in our study, due to saturation of the scattering detector"-> Was it not possible to utilize the low gain scattering detector? This should extend this upper limit further.

   **Authors Response:** We appreciate the reviewer's thoughtful comment. We would like to clarify that we utilized a low-gain channel for our analysis. We realize that our phrasing "due to saturation of the scattering detector" might have been unclear. Therefore, we have modified the sentence to enhance clarity as follows:

The upper coating thickness limit of LEO (line *ii*) was $\sim$ 285 nm in our study, due to saturation of the LSD before there is sufficient data below the 3% threshold of the maximum laser intensity to fit the leading edge

3) "morphological assumptions" -> please rephrase to "assumption regarding the morphology" or something like that.
**Authors Response:** Noted and revised the manuscript as:

To make a general comparison between the LEO results and those derived directly by the CPMA–SP2 measurements without making any assumptions regarding and density and morphology (Figure 2*b*),...

We believe that these revisions have improved the clarity of the manuscript. We hope that our responses and revisions adequately address your concerns and that you find the revised manuscript acceptable for publication in Journal of Atmospheric Measurement Techniques.

Thank you once again for your valuable feedback, which has helped us enhance the quality of our work. We look forward to your positive consideration of our revised manuscript.

---

## Author Response (AR2)

**Comparison of the LEO and CPMA-SP2 techniques for black-carbon mixing-state measurements**
**(Response to Reviewers)**

March 12, 2024

**Dear Reviewers,**

Thank you for providing us with your valuable feedback on our manuscript titled ***Comparison of the LEO and CPMA-SP2 techniques for black-carbon mixing-state measurements***, submitted to Atmospheric Measurement Techniques. We sincerely appreciate the time and effort you have dedicated to reviewing our work, and we are grateful for the opportunity to incorporate your constructive feedback into our revisions. After carefully considering all of your comments, we have made revisions throughout the manuscript to address them. To clearly indicate our responses to the reviewer's comments, we have highlighted them in red text text, while changes made to the manuscript are highlighted in blue text. We believe that these revisions significantly improve the quality and clarity of our manuscript. We are thankful for your valuable input, which has helped us enhance this work. We hope that our responses and revisions adequately address your concerns, and we eagerly await your positive consideration of our revised manuscript.

Response to Reviewers 2

**Comment:** There are still quite a few concerns of mine for this study.

**Authors Response:** We thank the reviewer for taking the time to do a second careful review, which raised several opportunities to further improve our manuscript. We trust that the reviewer will agree that we have done so.

**Comment:** The conclusion about the SP2-only method is not able to measure some smaller particles, which depends on a proper retrieval technique and needs more detailed discussions.

**Authors Response:** We have given a much more detailed description of our SP2-LEO analysis than is standard, and have cited this more extensively than in a typical SP2 paper. We have cited three different studies (Taylor et al., 2015; Zhao et al., 2020; Pileci et al., 2021) which are focussed on the detection limits of the SP2 and its implications, and one additional study which went into considerable detail on the topic (Dahlkötter et al., 2014). A repeated consideration of the detection limits of the SP2 would not be a novel contribution to the literature.

**Comment:** In addition, smoke may be the worst case for the comparison between both methods, because the particle is large and heavily coated.

**Authors Response:** We agree that large, heavily coated smoke particles (Case III) may not represent all atmospheric conditions. However, our Case I included small, thinly coated urban particles from a highway. So we believe we have covered a representative range of atmospheric conditions. Further studies will, of course, be valuable.

**Comment:** My main concern is about the results in Fig. 4, as the rBC measurement should look similar between both methods. The measurement efficiency of rBC by the SP2 has been well established, which should be nearly unit above 0.8fg. If this is the case, that means something wrong with the SP2 operation. If you have only shown the BCc with successful LEO retrieval (the retrieval fraction could depend on many things which is rather difficult to be discussed), which should be applied with a retrieval fraction rather than reporting only the uncorrected one.

**Authors Response:** There seems to be a misunderstanding here. Figure 4 does not show the SP2 measurement efficiency of rBC. The reviewer is probably looking at the distance from the dark blue line to the light blue line in Fig 4b – this is not an SP2 counting efficiency, but a LEO measurement efficiency. The reason that LEO did not succeed for many rBC-containing particles is that they were either too small or too thickly coated, i.e., they lay outside of lines (i) and (ii) in Figure 2b. We have modified Figure 4 and its caption to clarify this as follows:
  - Fig 4a shows the distribution of rBC particles as a function of total mass (not rBC mass-equivalent) according to the CPMA setpoint (CPMA-SP2) or light scattering (SP2-LEO).
  - Fig 4b shows the distribution of rBC mass which the reviewer was referring to. Here, we show that the SP2 incandescence only (pink line) and CPMA-SP2 data agree. We also show that the SP2-LEO method was only successful on a fraction of particles.
  - Fig 4c and 4d show Fig 4a and 4b weighted by mass.
We take responsibility for this miscommunication and have revised Fig. 4 to clarify these points.

[Figure]

(a)

(b)

(c)

(d)

We have also rewritten the Figure 4 caption:

Comparison of rBC−containing particle number and mass concentrations in Case II as functions of total particle mass (a and c) as functions of rBC-core mass (b and d). The figures shows data from both the LEO analysis and the CPMA-SP2 method. The detection-limit lines (iii) and (iv) from Figure 2 are reproduced in (b) and (d) here, to emphasize why the LEO results differ from the CPMA-SP2 results. The detection-limit line (i) from Figure 2 is the reason that the LEO data here drop to zero for small particles, relative to the CPMA-SP2 data. The pink dashed line in (b) illustrates the consistency between rBC distributions measured by CPMA-SP2 and SP2-only (standard measurement).

**Comment:** I am also a bit worried about Fig. 2 a and c, the LEO retrieval has been very different with CPMA-SP2 results. For example, how the SP2-only retrieval is so far from the lower limit dashline i, shouldn't it be close to the limit, otherwise it can't be deemed as a limit. There may be something wrong with the LEO fit. It would be very useful to check out if the coating/rBC ratio in bulk is consistent between both methods. This is important because even if SP2 is unable to detect them all, it can at least give the bulk information.

**Authors Response:** The Reviewer is correct that we could not have used the data in Fig 2a to define the lower limit (i). We did not. Instead, we calculated these lines from the light-scattering detector (LSD) and broadband incandescence detector (BID) limits of detection, as discussed on Page 13. To better clarify the meaning of these important lines, we have expanded our discussion on Page 13 to explicitly state the definitions of Lines i, ii, iii, and iv in words. We note that we took the concept for these Lines from Dahlkötter et al. (2014), which was cited in the text but not on Page 13. We have now added that missing citation:

Figure 2a and c demonstrate the relationships between rBC diameter and coating thickness derived from the CPMA−SP2 and the LEO analysis, respectively. The detection limits of the LEO method, inspired by the concept introduced in Dahlkötter et al. (2014), are indicated by red lines in Figure 2, and are defined as follows.

The reviewer's suggestion to investigate the coating/rBC ratio "in bulk between both methods" is an important one. This is the purpose of our Figure 4, which shows the overall size distributions for total particle mass (Fig 4a, 4c) and for rBC only (Fig 4b, 4d) from the two methods.

We note that even for the CPMA-SP2 method, a fraction of the smallest and largest rBC particles is not quantified; this has been discussed in detail by Pileci et al. (2021).

Thanks to the reviewer's comments, we added the above sentence to our discussion surrounding Figure 4d.

**Comment:** It looks CPMA-SP2 has a cut off at larger particle mass, which could be a limitation for such method but advantage for SP2-only. This should be discussed.

**Authors Response:** The CPMA is able to transmit larger particles than our SP2 was able to measure. This is an SP2 limitation (line (iv) in Figure 2b) and not a CPMA limitation. We therefore discussed this implicitly as appropriate. Nevertheless, the Reviewer's point is important, and was addressed by the new citation to Pileci et al. (2021) above.

**Comment:** I can't see the point to show Fig. 3 as they all look similar to Fig. 2. This should be at least mentioned in the figure caption. How much uncertainty when deriving from the uncoated and coated BC mass distribution from the CPMA-SP2, considering it is a 2D inversion to get this information. This should be discussed in addition to the disadvantage of the SP2-only method.

**Authors Response:** The reviewer is correct that we duplicated Fig 2b and 2d in Fig 3b and 3e. This was to show the full range of our data set. However, we reconsidered following this comment.

Indeed, we can omit Fig 3b and 3e (Case II) since the other panels in Fig 3 show the extreme minimum (Case I) and maximum (Case III) of wildfire smoke influence. We revised the figure. We also revised the caption of Fig. 3 to be clearer and briefer.

[Figure]

Figure 1: Distributions of number concentration for Cases I (clear visibility, thinnest rBC coatings, urban influence) and III (poorest visibility, thickest rBC coatings, wildfire influence), similar to Figure 2b and Figure 2d (Cases II). Case I represents the smallest influence of wildfires observed in our study; Case III represents the maximum. Solid diagonal lines indicate pure rBC particles with an rBC mass fraction of unity ($m_{\mathrm{rBC}}/m_{\mathrm{p}} = 1$), while parallel lines represent decreasing mass fractions of 0.75, 0.5, and 0.25, respectively.

**Other changes to the manuscript:** We revised some of the text and figure captions for clarity. We revised Figure 1 to show collapsed soot inside the mixed particles, according to recent work (Corbin, Modini, and Gysel-Beer, Aerosol Sci Technol., 2023, link to paper).

[Figure]

We are confident that these revisions further enhance the clarity and quality of our manuscript. We trust that our responses and modifications effectively address the issues raised, rendering the revised manuscript suitable for publication in the Journal of Atmospheric Measurement Techniques.

Thank you once again for your time and attention to our study.

We believe that these revisions have improved the clarity of the manuscript. We hope that our responses and revisions adequately address your concerns and that you find the revised manuscript acceptable for publication in Journal of Atmospheric Measurement Techniques.

Thank you once again for your valuable feedback, which has helped us enhance the quality of our work. We look forward to your positive consideration of our revised manuscript.